

# Keep the ball rolling: sexual differences in conglobation behavior of a terrestrial isopod under different degrees of perceived predation pressure

Francisco Javier Zamora-Camacho

Department of Biogeography and Global Change, Museo Nacional de Ciencias Naturales, Madrid, Spain

## ABSTRACT

**Background**. Antipredator behaviors are theoretically subjected to a balance by which their display should be minimized when their benefits do not outweigh their costs. Such costs may be not only energetic, but also entail a reduction in the time available for other fitness-enhancing behaviors. However, these behaviors are only beneficial under predation risk. Therefore, antipredator behaviors are predicted to be maximized under strong predation risk. Moreover, predation pressure can differ among individuals according to traits such as sex or body size, if these traits increase vulnerability. Antipredator behaviors are expected to be maximized in individuals whose traits make them more conspicuous to predators. However, how sex, body size and antipredator behaviors interact is not always understood.

**Methods**. In this work, I tested the interaction between sex, body size and antipredator behavior in the common pill woodlouse (*Armadillidium vulgare*), which conglobate (*i.e.,* they roll up their bodies almost conforming a sphere that conceals their appendages) in response to predator attacks. Specifically, I tested whether latency to unroll after a standardized mechanical induction was greater in animals exposed to predator chemical cues (toad feces) than in conspecifics exposed to cues of non-predatory animals (rabbits) or no chemical cues whatsoever (distilled water), incorporating sex and body mass in the analyses.

**Results**. In agreement with my prediction, latency to unroll was greater in individuals exposed to predator chemical cues. In other words, these animals engage in conglobation for longer under perceived predator vicinity. However, this result was only true for males. This sexual dimorphism in antipredator behavior could result from males being under greater predation risk than females, thus having evolved more refined antipredator strategies. Indeed, males of this species are known to actively search for females, which makes them more prone to superficial ground mobility, and likely to being detected by predators. Body size was unrelated to latency to unroll. As a whole, these results support the hypothesis that antipredator behavior is tuned to predator cues in a way consistent with a balance between costs and benefits, which might differ between the sexes.

Corresponding author
Francisco Javier Zamora-Camacho, zamcam@ugr.es

## INTRODUCTION

Predators erode their prey's fitness in various ways, thus embodying a potent selective pressure on them (*Abrams, 2000*; *Lima, 2002*). First and foremost, successful predatory events involve the annihilation of the prey's life, and consequently of any potential future fitness it might have had (*Barbosa & Castellanos, 2005*; *Beauchamp, Wahl & Johnson, 2007*). However, predators also exert non-lethal effects on their prey that are also pivotal in multifarious ways (*Lima, 1998*; *Preisser, Bolnick & Benard, 2005*; *Wirsing et al., 2021*). After consumption, the second gravest damage predators inflict on their prey is probably represented by physical injury following failed attacks (*Laha & Mattingly, 2007*; *Bowerman, Johnson & Bowerman, 2010*), which frequently entail infections (*Aeby & Santavy, 2006*) as well as impaired locomotion, growth, and ultimately fitness (*Archie, 2013*; *Zamora-Camacho & Aragón, 2019*; *Zamora-Camacho & Calsbeek, 2022*). Even in the absence of an actual attack, preys are bound to face the harmful effects of predators. Some animal species innately possess physical (*Mukherjee & Heithaus, 2013*) or chemical defenses (*Glendinning, 2007*), occasionally remarkably sophisticated (*Zamora-Camacho, 2023*), which can dissuade predators (*Brown et al., 2016*). Moreover, most prey are equipped with sensory systems capable of detecting predator vicinity (*Leavell & Bernal, 2019*). Such perceived predator proximity oftentimes elicits the expression of inducible morphological or chemical defenses of different types (*Kishida et al., 2010*; *Yamamichi et al., 2019*). In either case, whether innate or inducible, these defenses can be costly, given the energy diverted to their production (*Hamill, Rogers & Beckerman, 2008*; *Gilbert, 2013*; *Hermann & Thaler, 2014*; *Zvereva et al., 2017*), and the fact that the metabolic processes involved in these responses may even trigger oxidative stress (*Janssens & Stoks, 2013*).

On a different note, prey can also tune their behavior to the threat represented by potential predators (*Lima & Dill, 1990*; *McGhee, Pintor & Bell, 2013*) and the level of risk involved (*Kavaliers & Choleris, 2001*). The most immediate antipredator behavior is oftentimes spatial circumvention, which prevents an actual encounter (*Palmer et al., 2022*; *Suraci et al., 2022*). Also, prey typically respond to predator proximity by diminishing the conspicuousness of their activities (*Moll et al., 2020*; *Balaban-Feld et al., 2022*). When the encounter is imminent, however, prey can decide whether to face or avoid the predator depending on the chances of success of each strategy (*Reichmuth et al., 2011*; *Zhang et al., 2020*). A particularly common reaction of prey to such encounters is flight (*Møller & Erritzøe, 2014*; *Basille et al., 2015*). In addition, more refined behaviors against predation are likewise common, such as postural strategies that facilitate the deflection of the attack towards a non-vital (*Myette, Hossie & Murray, 2019*) or well protected body region (*Crofts & Stankowich, 2021*), that make it difficult for the predator to handle and subdue the prey (*Kowalski, Sawościanik & Rychlik, 2018*), or that invoke death feigning or thanatosis (*Humphreys & Ruxton, 2018*). This wide array of antipredator behaviors can coexist in the same individual and be subjected to complex interactions (*Lind & Cresswell, 2005*).

In any case, antipredator behavior is not devoid of costs. Besides the energy demands of strategies such as flight, which involves a frequently intense muscular exertion (*Biewener & Patek, 2018*), a cost in terms of fitness is expected given that antipredator behaviors

are time-consuming (*Lima & Dill, 1990*) and thus reduce the time devoted to foraging, mating and reproducing (*Langerhans, 2007*; *Gulsby et al., 2018*). The final decision of a prey regarding whether and to which extent to engage in antipredator behaviors should be made considering a balance between their costs and benefits (*Herberholz & Marquart, 2012*). Indeed, antipredator defenses are expected to be selected against in the absence of predators (*Reznick, Ghalambor & Crooks, 2008*; *Palkovacs, Wasserman & Kinnison, 2011*), at least to a certain extent (*Blumstein, 2006*), which could release the bearer from the costs associated to such behaviors if they are no longer beneficial. In fact, the success of a given antipredator behavior depends on diverse circumstances, and can vary according to predatory pressure and the qualitative and quantitative expressions of the antipredator behaviors adopted by other potential preys (*Menezes, 2021*).

Indeed, prey are predicted to adjust their antipredator behavior to the actual intensity of predator pressure, responding strongly when predators are an actual threat, but mildly when that threat is lesser (*Sih, Ziemba & Harding, 2000*; *Ferrari, Sih & Chivers, 2009*). Also, even at the intraspecific level, some individuals can be at higher risk than others, depending on differences in morphology (*Zamora-Camacho, 2022*) and personality (*Sommer & Schmitz, 2020*) that can make some individuals more or less prone to succumb to predator attacks. Given that, probably as a part of their mating strategies, males are often morphologically (*Williams & Carroll, 2009*) or behaviorally (*Schuett, Tregenza & Dall, 2010*) more conspicuous than females, males can be subjected to a stronger predation pressure than females (*Husak et al., 2006*; *Kojima et al., 2014*), thus responding with stronger antipredator strategies (*Husak & Fox, 2008*; *Zamora-Camacho, 2022*).

In this context, this work aims to contextualize the display of an unusual antipredator behavior, conglobation in common pill woodlice (*Armadillidium vulgare*), as a function of extrinsic factors, such as predator cues, and intrinsic traits, such as body mass and sex, which relationships are poorly understood. Conglobation is a particular behavior by which these animals coil up into a ball when disturbed, concealing their appendages within their dark grey cuticle (*Cazzolla Gatti et al., 2020*). This position makes them not only difficult to handle, but also resemble a pebble rather than edible animals (*Tuf & Ďurajková, 2022*), which has been interpreted as tonic immobility or even as thanatosis (*Horváth et al., 2019*; *Cazzolla Gatti et al., 2020*). Therefore, this behavior can be particularly efficient against non-gape limited predators, especially those which detect their prey through their movements, such as amphibians. Specifically, I studied the time spent by male and female *A. vulgare* in the conglobated position in the presence and the absence of olfactory predator (toads) cues after conglobation was mechanically induced in a standardized way (poking the animals with a stick), using chemical cues of non-predatory animals (rabbits) as well as no odor as controls. In line with the aforementioned rationale that antipredator behavior is costly, I predict that the conglobated position will be abandoned earlier in the absence of predator cues, when its potential benefits are lower. Also, I expect that, if one of the sexes is under greater predation pressure (which might be the case of males, which seem to be more active according to certain evidence *Dangerfield & Hassall, 1994*), this risk will have selected for a stronger reaction to predator cues.

## MATERIALS & METHODS

### Study species

The common pill woodlouse (*A. vulgare*) is a terrestrial isopod, native to the Mediterranean region but introduced worldwide (*Schmalfuss, 2003*), that occupies a variety of temperate habitats. It shows a preference for a certain degree of moisture (*Bonuti et al., 2021*), which can determine some extent of small-scale seasonal migration in search of sufficient yet not excessive humidity (*Paris, 1963*). Reproduction takes place in the summer in cold regions (*Dangerfield & Hassall, 1992*), but in spring in more temperate areas (*Sorensen & Burkett, 1977*). Females possess a ventral marsupium where eggs are deposited until hatching (*Suzuki, 2001*; *Suzuki & Futami, 2018*). As a macrodecomposer, it feeds on a variety of dead organic matter sources (*Paris, 1963*) which it selects according to its quality (*Tuck & Hassall, 2005*). In turn, a wide array of invertebrates, amphibians and reptiles have been cited as predators of this species (*Paris, 1963*). Against these predators, *A. vulgare* can resort to numerous morphological and behavioral defenses, such as crypsis, immobility, escape or sheltering, among which conglobations is particularly common (*Horváth et al., 2019*). However, males could be more active than females (*Dangerfield & Hassall, 1994*), which might lead to greater predation pressure (*Yli-Renko, Pettay & Vesakoski, 2018*), with the concomitant sexual divergence in antipredator responses and success (*Yli-Renko, Vesakoski & Pettai, 2014*). Also, larger individuals tend to take greater risks in this species (*Horváth et al., 2019*).

### Animal capture and management

Fieldwork took place in Pinares de Cartaya (SW Spain; 37° 21′ N, 7° 11′ O), an 11,000-ha *Pinus pinea* grove with an undergrowth dominated by *Rosmarinus officinalis*, *Pistacia lentiscus* and *Cistus ladanifer*. In this forest, I collected 43 adult *A. vulgare* (19 females and 24 males) by hand, searching under rocks, decaying logs, and other potential refugia at appropriate sites. However, to diminish the chances of capturing genetically related individuals, only one specimen was caught at a given site, and at least 50 m were left among sites (*Horváth et al., 2019*; *Beveridge et al., 2022*). Sampling took place in February 2022, immediately before the onset of the mating season (which beings in the early spring in this area, pers. obs.), because parental care can affect antipredator behavior in females, involving a difficulty in the adoption of the conglobated position, which could affect the results (*Suzuki & Futami, 2018*).

The animals captured were transferred to the laboratory, where they were assigned an ID number, weighed to the nearest 0.01 g with a CDS-100 scale, and individually housed in cylindric plastic terraria (6 cm diameter × 15 cm height) with wet peat as a substrate, a piece of fresh carrot as nourishment, and a wet cotton disk (4 cm diameter × 1 mm thick) above it as a shelter. The terraria were randomly set in a shelve in the laboratory, and their position was changed every 24 h. A window let daylight in, which permitted the adjustment of circadian rhythms. Room temperature was not manipulated, and fluctuated naturally between 10 °C at night and 20 °C during the day.

The behavioral tests began 24 h after capture. These tests were conducted in individual cylindric plastic terraria (4 cm diameter × 10 cm height) with a cotton disk lining (4 cm

diameter × 1 mm thick) at the bottom. This species interprets chemical cues to identify dead conspecifics (*Yao et al., 2009*), potential mates (*Beauché & Richard, 2013*) and predators (*Pniewski, 2014*), and tunes its conglobation behavior to diverse environmental factors (*Horváth et al., 2019*). Therefore, I used different chemical cues (or the absence thereof) in three separate tests. In the experimental tests, the cotton disk at the bottom of the terrarium was soaked with a 1-mL aliquot extracted from a preparation of 0.5 L of distilled water where 50 g of a mix of fresh feces from two male and two female adult common toads (*Bufo spinosus*), captured in the same habitat as the woodlice, had been diluted. These toads are abundant and widespread generalist predators of invertebrates, including isopods (*Ortiz-Santaliestra, 2014*). In the control tests, the cotton disk at the bottom of the terrarium was soaked with a 1-mL aliquot extracted from a preparation of 0.5 L of distilled water where 50 g of a mix of fresh feces from four different European rabbit (*Oryctolagus cuniculus*) latrines (separated by at least 600 m) from the same habitat as the woodlice, had been diluted. These rabbits are abundant and widespread generalist herbivores (*Gálvez-Bravo, 2017*). Feces of both toads and rabbits, these originated from natural, uncontrolled diets, thus representing what the isopods are likely to find in nature. In the manipulation control tests, the cotton disk at the bottom of the terrarium was soaked with one mL of distilled water. In this way, humidity was constant across tests, which avoided a potential effect of moisture on conglobation behavior, as conglobation can also be a behavioral strategy against water loss in these animals (*Smigel & Gibbs, 2008*).

For these tests, each individual was placed alone in one arena as described above. After 5 min for habituation, I gently poked the animal with a wooden stick until it adopted the fully conglobated position. The test ended when the individual abandoned this position. All individuals underwent all three tests, with a 24-h resting period in between. Every time, the cotton disks were replaced and the arenas were rinsed thoroughly. Conglobation behavior in these animals is affected by previous experience (*Matsuno & Moriyama, 2012*). For that reason, the sequence in which the tests involving the different stimuli were conducted was random for each individual.

All tests were recorded with a Canon EOS 550D video camera. The resulting footages were then studied using the software Tracker v 6.0.8, which allows frame-by-frame analyses. Specifically, I measured latency to unroll as the time each individual spent in the conglobated position, by recording the time elapsed since the frame in which this position was adopted until it was abandoned. After the tests, the woodlice were sexed, based on the presence of the marsupium in the ventral side of the pereion in females after the parturial mold prior to reproduction (*Surbida & Wright, 2001*; *Suzuki, 2002*), and released in the same habitat where they had been captured.

## Statistics

Latency to unroll needed to be ln-transformed in order to meet the assumptions of homoscedasticity and residual normality needed for parametric statistics (*Quinn & Keough, 2002*). After that, a mixed model was conducted where latency to unroll (ln-transformed) was the response variable, sex, treatment and their interactions were included as factors, body mass was included as a covariate, and ID was a random factor. Sum of squares

**Table 1  Tukey post-hoc test performed on the Sex × Treatment interaction.** *t*- and *P*-values for each pairwise comparison are indicated. Significant results are in bold.

| Pairwise comparison | *t*-value | *P*-value |
| --- | --- | --- |
| Female Rabbit vs Male Rabbit | 1.213 | 0.830 |
| Female Rabbit vs Female Toad | 0.376 | 0.999 |
| Female Rabbit vs Male Toad | −1.414 | 0.719 |
| Female Rabbit vs Female Water | 1.722 | 0.521 |
| Female Rabbit vs Male Water | 1.646 | 0.570 |
| Male Rabbit vs Female Toad | −0.856 | 0.956 |
| **Male Rabbit vs Male Toad** | **−3.107** | **0.030** |
| Male Rabbit vs Female Water | 0.424 | 0.998 |
| Male Rabbit vs Male Water | 0.512 | 0.996 |
| Female Toad vs Male Toad | −1.771 | 0.489 |
| Female Toad vs Female Water | 1.347 | 0.758 |
| Female Toad vs Male Water | 1.289 | 0.791 |
| **Male Toad vs Female Water** | **3.050** | **0.033** |
| **Male Toad vs Male Water** | **3.619** | **0.007** |
| Female Water vs Male Water | 0.009 | 1.000 |

was type III. A Tukey *post-hoc* test was applied on the interaction term. These tests were conducted with the package *lmerTest* (*Kuznetsova, Brockhoff & Christensen, 2017*) in the software R *v.* 4.1.2 (*R Core Team, 2021*). A similar test but excluding sex can be found as Supplementary Material.

## RESULTS

Body mass had no significant effect on latency to unroll ($F_{1,122} = 0.698$; $\beta = -2.843$; $P = 0.409$). The effect of sex on latency to unroll was non-significant ($F_{1,122} = 0.073$; $P = 0.789$), but that of treatment was significant ($F_{2,122} = 5.823$; $P = 0.004$). According to the Tukey post-hoc test applied on the marginally non-significant Sex × Treatment interaction ($F_{2,122} = 2.786$; $P = 0.068$), males exposed to toad scent had greater latency to unroll than males exposed to rabbit scent and to water, and than females exposed to water, with every other pairwise comparison being non-significant (Table 1; Fig. 1). When sex was excluded from the model, treatment had a significant effect on latency to unroll, where the only significant pairwise comparison was between the treatments with water and toad cues according to the Tukey *post-hoc* test (Supplementary Material).

## DISCUSSION

Some of these results were in agreement with my predictions. In the first place, latency to unroll was greater in the presence of predator chemical cues than in the absence of it. According to theory, predator vicinity can trigger a fear response on the prey, which is not devoid of costs (*Wang & Zoy, 2018*; *Qiao et al., 2019*; *Tripathi et al., 2022*). Previous research supports that, in behavioral terms, most prey reduce their susceptibility to predators by diminishing their activity rates when threatened (*Brodin & Johansson,*

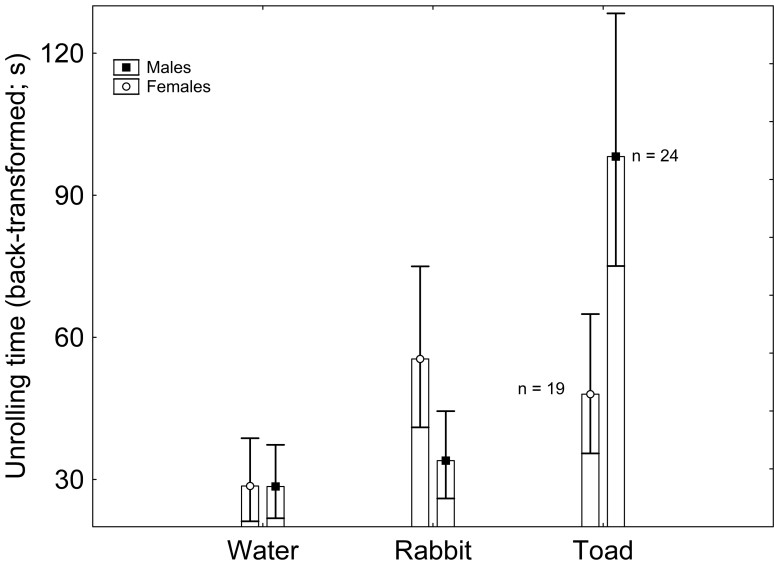

**Figure 1** **Sex and treatment differences in unrolling time (back-transformed).** Vertical whiskers represent standard errors. Sample sizes are indicated.

*2004*; *Laurila, Pakkasmaa & Merilä, 2006*), even resorting to total immobility (*Brooks, Gaskell & Maltby, 2009*) and death feigning (*Konishi et al., 2020*). However, by engaging in such antipredator behavior, prey inevitably reduce the amount of time available for other fitness-enhancing activities, such as mating, feeding, and territory defense (*Persons, Walker & Rypstra, 2002*; *Lind & Cresswell, 2005*), which may entail negative effects, for example on growth (*Brodin & Johansson, 2004*; *Laurila, Pakkasmaa & Merilä, 2006*) and reproduction (*Persons, Walker & Rypstra, 2002*; *Kempraj, Park & Taylor, 2020*). These costs can be assumed to affect *A. vulgare* when remaining in a conglobated position, although little is known in this regard about this particular species. Thus, such antipredator behaviors are allegedly subjected to a balance between these costs and their benefits, namely predator avoidance. In this context, prey are expected to minimize antipredator behaviors when their benefits are scarce, *i.e.,* under low predation risk (*Ferrari, Messier & Chivers, 2008*; *Supekar & Gramapurohit, 2020*; *Batabyal et al., 2022*). This prediction is supported by these results, as latency to unroll was greater in the presence of predator chemical cues presented in the short term. Similarly, the marine isopod *Idotea balthica* lowers its activity in the presence of chemical cues from a native predatory fish (*Yli-Renko et al., 2022*). However, a different study reports that *A. vulgare* remains unresponsive to chemical cues of an arachnid predator (*Zimmerman & Kight, 2016*).

Nonetheless, this greater latency to unroll in the presence of predator chemical cues was only observed in males, whereas females did not respond to these cues with an increase in time to unroll. This observation is based on an interaction between sex and treatment that was marginally non-significant, but it provides a hint of sex differences in responses to treatments. While the possibility that females lack the ability to recognize predator chemical cues cannot be discarded, a greater response of males as a result of a male-biased

predation risk could be a more plausible explanation. In circumstances where both sexes are under equivalent risk, their response to predator cues might not differ (*David, Salignon & Perrot-Minnot, 2014*; *Kempraj, Park & Taylor, 2020*; *Saavedra, Tomás & Amo, 2022*). However, whenever one sex is under greater risk than the other, it is expected to evolve more efficient antipredator responses (*Curio, Klump & Regelmann, 1983*). Although in some species females have been found to face greater predation risk (*Post & Götmark, 2006*) and to respond with greater intensity to predator pressure (*Pärssinen et al., 2021*; *Woodrow et al., 2021*), in most cases males are more conspicuous to predators as a result of more active behaviors (*Tobler, Franssen & Plath, 2008*), such as territory defense (*Gwynne & O'Neill, 1980*), female pursuit (*Fišer et al., 2019*) and courtship (*Whitaker et al., 2021*). Accordingly, males display a stronger behavioral response to predation risk in taxa as disparate as mammals (*Grignolio et al., 2019*), birds (*van den Bemt, steves Lopes & Ribeiro Cunha, 2021*), reptiles (*Bohórquez Alonso et al., 2010*), snails (*Donelan & Trussell, 2020*), insects (*Schultz, 1981*), spiders (*Krupa & Sih, 1998*) or crabs (*Jennions et al., 2003*).

In the specific case of *A. vulgare*, sexual divergence in activity has yet to be studied, but different lines of evidence suggest that males could be more active, and thus more detectable by predators, which could favor a greater investment in antipredator behavior. In the first place, genetic analyses have revealed that females are philopatric whereas males are not, which is compatible with males being more prone to dispersal and, allegedly, to be intercepted by predators (*Durand et al., 2019*). Moreover, males are known to actively search for females based on chemical cues (*Beauché & Richard, 2013*) and to compete for access to them given their multiple paternity scheme (*Verne et al., 2007*; *Valette et al., 2016*). Also, male presence can stimulate female receptiveness (*Lefebvre & Caubet, 1999*). These features could be accompanied by behavioral displays that might increase male conspicuousness to predators. Indeed, males could be more active in the ground surface, whereas females tend to make a greater use of underground shelters, which is a probable consequence of the former actively competing and searching for the latter (*Dangerfield & Hassall, 1994*). Nonetheless, until all of these facts are properly studied, this assumption can be considered plausible, but speculative.

In correspondence with previous studies on this species (*Beveridge et al., 2022*), body mass was uncorrelated with latency to unroll, as well as with other antipredator behaviors (*Cazzolla Gatti et al., 2020*). This finding contrasts with research that indicates that antipredator behavior depends on body size on other taxa, both vertebrates (*Hoare et al., 2000*; *Roth & Johnson, 2004*) and invertebrates (*Johnson et al., 2017*; *Gavini, Quintero & Tadey, 2020*), including larger crustaceans (*Wahle, 1992*). In this case, the relatively small size of the focal species might make variation in body size irrelevant for most potential predators, thus not selecting for differential antipredator strategies at varying sizes. In any case, conglobation behavior is known to be repeatable in this species (*Cornwell et al., 2023*), which advocates for consistency in the patterns described herein.

## CONCLUSIONS

To conclude, latency to unroll was greater in individuals exposed to predator chemical cues, which supports the prediction that *A. vulgare* can detect these cues and react accordingly,

although these differences were led by males. Moreover, these findings (selection for antipredator responses is stronger in males) concurs with the theoretical assumption that antipredator behaviors are subjected to a cost-benefit balance, by which they should be minimized when their benefits do not outweigh their costs. This supports the prediction that males are under greater predation risk than females, thus having evolved more refined antipredator strategies, and that there is a cost implicit in conglobation behavior that females avoid paying by not responding to the same stimulus in the same way males do. Antipredator behaviors are only beneficial under predation risk, which could be the reason why males engage in conglobation for longer under perceived predator vicinity.

## ACKNOWLEDGEMENTS

Comments by Ivan Tuf and Outi Vesakoski improved the manuscript.

### Funding
The author received no funding for this work.

### Competing Interests
The author declares that he has no competing interests.

### Author Contributions
- Francisco Javier Zamora-Camacho conceived and designed the experiments, performed the experiments, analyzed the data, prepared figures and/or tables, authored or reviewed drafts of the article, and approved the final draft.

### Data Availability
The data are available in the Supplemental Files.

### Supplemental Information
Supplemental information for this article can be found online at http://dx.doi.org/10.7717/peerj.16696#supplemental-information.

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
