# Peer review of "Keep the ball rolling: sexual differences in conglobation behavior of a terrestrial isopod under different degrees of perceived predation pressure"

_PeerJ, doi:10.7717/peerj.16696_

## Round 0.1 · original submission · Major Revisions

Please revise your manuscript to address all of the concerns of the reviewers. Note that it will be re-sent to the reviewers.

·

Basic reporting

The literature references are numerous (it is possible to delete some used to illustrate ecological/evolutionary/behavioural concepts in unrelated taxa).
The lines (solid, dashed) should be omitted from Figure 1 as the mean values are unrelated (no order between treatments, no timing, no changes) - despite the confusing lines, different symbols for males/females should be used. (Nevertheless, these 6 points can be presented in the table in a more space-saving way.)
The raw data are not shared.

Experimental design

The missing/confusing part of the methods is the collection of individuals. The timing is too early to associate with the breeding season (if author disagrees, he should provide information on the timing of the presence of gravid females in SW Spain). Also, Armadillidium vulgare uses different antipredatory strategies (colouration, immobility, running away, burrowing/sheltering, …), you should explain why you assume that conglobation can be useful as a strategy against toad (there are numerous references).
The sexing of isopods (lines 174-175) is meaningless because females do not have abdominal pouches. The marsupia are located on the ventral side of the pereion (Crustacea/Isopoda do not have an abdomen but a pleon), yet the marsupia are for the storage of eggs and mancas EXCLUSIVELY. This means that the presence of marsupia is related to gravidity. (The author sampled the isopods in February to avoid gravid females.)

Validity of the findings

The differences are small, the results supporting the first objective are presented only as F and P values (effect of treatment). The “raw” length of latency to unrolling should also be presented for all individuals together (Fig. 1 shows only ln-transformed values for each sex).
The result of the effect of ant odour on isopod behaviour (lines 213-214) is not closely related to the results presented. The toad is a predator that can swallow the whole isopod (if detected) immediately, while the ants have to attack the isopod in a group and conglobation is a pretty good strategy to avoid their attack. This means that the short-term “stress” of the ant(s) is low (they just walk around), but the long-term stress is related to a (new) ant nest nearby. Such a comparison with toads is not useful.
However, the author missed a paper on the effect of chemical cue of predators on isopods (DOI: 10.9784/LEB4(2)Zimmerman.01) – Armadillidium did not react to the smell of the spider. This article describes well the strength of the effect of predator odour on prey as a function of the predator's diet. There is also an article on isopod avoidance of dead conspecifics (DOI 10.1007/s11692-009-9069-4), so the author should pay attention to this – are faeces from toads fed on isopods more effective for latency to unroll than faeces from toads fed on other prey?
The author´s assumption that males are subjected to higher predation pressure is weak. In the first – among isopods there is promiscuity (https://doi.org/10.1016/j.beproc.2019.104030), so it is not necessary to be the first mate of the female (= to compete with other males).
The author´s conjecture about males seeking out females is based on Dangerfield and Hassall (1994), but their interpretation is based on raw (fluctuating) data (cf. 4 lines in Fig. 1) without any statistical analysis. In any case, the higher proportion of males in shelters says nothing about their activity on the surface, nor about the surface activity of females and its timing. The second aim of the presented manuscript is too speculative.

·

Basic reporting

(Subtitles are achieved from the instructions to the reviewers)

This work investigates if antipredator response
1) take place in woodlice (natural enemy vs 2 controls, which is nice set-up)
2) is different between sexes
3) vary according to the size of the woodlice
It discusses the differences in antipredator responses in evolutionary framework considering costs and benefits of the behavior.

* Language questions *

Personal statement on language policy: As none native speaker of English, I try to promote equality between scientists with various linguistic origin and do not assume perfect English from anyone. Not that I would recognize it if I would see one. This is also to say that my tolerance to non-standard English and ability to parse the non-standard forms is high.
Comment on the language of this paper: It is totally great.

* Context given in intro & literature review *

Abstract “background” gave lots of context, but the author could state still that “what information/proof is missing from the word” = why this study is important to do. Methods do not mention what is the option to predator chemical cue; how was the conglobation induced? With chemical cue and with what?

- In line 162 we learn that the response was induced by poking the animal. This could be mentioned to make the set-up more clear already here. Also in Aims (last paragraph of Introduction).
- In line 147 we learn that there were cue of toads (predator) and rabbits (random fellow from the same environment) and control water (to control for the manipulation. This is a solid approach, and it could be mentioned in abstract and also in the Aims (last paragraph of Introduction).

Introduction provides a coherent background for types of antipredatory responses and their evolution balancing costs and benefits. Only the lines 80-83 are not clear to me. The background for the study is clear, but I am missing the paragraph that says what is missing from the ecological knowledge in regards to inter-individual variation in “inducible” antipredator responses. The line 94 states that this work aims to “contextualize” the display of an unusual antipredator behavior –” It could be maybe put more “aggressively”, that “this and that is not known and thus I studied the inter-individual variation of latency of conglobation behavior of a woodlice.”

Aims continues: It would be good to know what triggers the conglobation in the study. It is not clear for naiive reader that why would the fellow be in conglobated position WITHOUT a hint of predator around. Also, the author expects that if one of the sexes is under greater predation pressure, there should be differences in the outcome. It could be mentioned that IS there a sex-difference in the predation pressure in this species and does this fact make the author assumes that males should have stronger reaction (=one or two-sided hypothesis?).

- Later in Discussion we learn that there are something known of the sex differences. Please provide something from this to the Introduction to act as background of why to study this in this species.

Aims: In Methods, we learn that there were “different chemical cues (and the absence of thereof) in three separate tests” and that they were toad, rabbit as control and distilled water as treatment control. Please tell this info earlier!

The first paragraph of methods explain the ecology of the woodlice. The description is pretty general, and I would expect to hear what is known about its antipredator adaptations and sexual differences in behavior & antipred adaptation. Further, please provide information of factors that affect its size variation. And if nothing is known, that could be mentioned as well. At the moment it is unclear how the ecology of the animal allows its usage as study object for this study. (In lines 144-146 and 231-242 we do hear about these issues! It would be good to hear about those already in the first paragraph, and in length.)

* Structure following to PeerJ standards *

I don’t fully understand what is the “Background” section PeerJ wants to have after the Abstract. Now there is the same text in “Background” and “Abstract”. This is easily changed if the idea is to have different material in the two sections.

* Figures*

Fig. 1: It would be great to get the figure with back-transformed response variable. Then the mean values would make more sense.

Table 1 about the Sex * Treatment interaction: With multiple comparisons we are facing the problem of adjusting significance levels. On the other hand, not all comparisons are relevant. How about running “contrast-statements”, i.e. pairwise comparisons only for the relevant ones?:
male_toad – male_rabbit , male_toad – male_water and male_water – male_rabbit (variation within male responses)
same for ladies (variation within female responses)
male_toad- female_toad, male_rabbit – female_rabbit, male_water -female_water (variation within per treatment responses)

* Raw data supplied *

The data is provided, but something I was missing, please see below.

Experimental design

* Original research? *

We made pretty similar set-up (3 treatments, males and females, size) in marine isopod, that was published last year. This is not mentioned in the present ms but maybe it could be added to the discussion. Also in this case the response was more clear in males. The current ms ends the Discussion to hint for personality variation between individual, which we had as part of the study set-up. Also this angle could be interesting to the current ms.

Yli-Renko, M., Pettay, J., Rothäusler, E. & Vesakoski, O. Lack of anti-predator recognition in a marine isopod under the threat of an invasive predatory crab. Biological Invasions, 24: 3189–3198. https://doi.org/10.1007/s10530-022-02839-x

* Questions well defined, relevant and meaningful? *

As mentioned above, I would like to see more clearly defined statement of what is missing from our knowledge -> thus this study.

* Technical and ethical standards?*

Could you also test the first trial results only? (Given that you randomized the treatments within the individuals.) You mention, that the animal “learns” the set-up and respond in different way in the subsequent trials. This indeed seem to happen in many settings, and it is good that you report it. We cannot really trust to get a “natural” response in the first trial – OR is that natural or is it a response of overtly stressed animal? This is an important point for future repeated measures set-ups, and please feel free to write a paragraph of the phenomenon to the discussion for other to refer and discuss.

Please provide the pairwise comparisons of Treatment-factor and the mean values + errors of the “water”, “rabbit” and “toad”. Maybe there were no significant comparisons, but that would be ok too. At the moment it is difficult to assess if lines 196-213 follow from Treatment or Sex-Treatment factors.

Female high response to rabbit cue is unexpected but also not significant. (If all the animals got "rabbit" treatment first, this could be explained by order of treatments.)

I am fine with reporting the pairwise comparisons with the interaction p-value of 0,068. However, something in this style could be stated “The interaction was only marginally significant, but provides a hint/cue/whatever of sex differences in responses to the treatments. I wanted to study this further and run the pairwise comparisons… “ And in discussion it could be stated that the N was pretty low for finding stronger comparisons. Of course some power tests could give an idea whether we could assume to have a more clear interaction if N was higher.

* Methods properly described *
Yes.

Validity of the findings

* Validity of findings *

The biggest problem is that the pairwise comparisons within the 3 Treaments are not provided. Thus, it is difficult to say if the first section of the discussion is solid – was the latency higher in “toad” treatment than other or did we get the significance only in the pairwise comparisons of the interaction term (where male_toad and male_rabbit became significant). This is easily remedied, however.

The first paragraph of Discussion further discuss balancing between benefits and costs of antipredator behavior. Howeve, it is not clear if the woodlice in question has any costs of not unrolling fast. What do we know about the costs? If there are no known costs, do we have a case where we could assess the cost and benefit of this precise behavior?

In lines 214-230 we discuss about sex-differences in the predation risk and many other behaviors. The sex differences are offered as the reason for males responding to toads but females not. However, this logic would necessitate information of sex-biased behavior or predation risk in the woodlice. What is known? If nothing, then we cannot state that the reason is in stronger evolutionary pressure on male antipredator adatations (through higher predation risk).

BUT: Further lines (231 onwards) provide the woodlice-specific information that we needed! Could you lift this info to the 1st paragraph of the methods? It indicates, that we could indeed have a case where to expect sex differences in antipred adaptations. Please also add some prior information of body size to methods and tell why it was included to the model.

* Impact and novelty *

Antipredator responses are still studied a lot, but evolutionary dynamics of them is still partly unknown. This study is part of the cumulative sciences in this field.

As mentioned above, we made a pretty similar study earlier, but also there the sex-difference in responses were only marginal. Thus the question did got solved.

* All data provided and of good quality *

The provided data does not have information of the order the individuals achieved the treatments. If all the animals got the “rabbit” cue in the first trial, then we would actually assume stronger response to that than to the others. I hope that the treatments were given in randomized order.

* Conclusion well stated? *

I already commented the Discussion, and here I only refer to the chapter Conclusions.

255-256: Please add something in style of “latency to unrolla was greater tin individuals exposed to predoar chemical cues THAN TO CONTROL TREATMENT, BUT ONLY IN MALES”.
257-258: More information on the costs (is remaining in conglobation costly?) and benefits (in this species what you miss by not unrolling?) are needed before this statement. See comments above for this.

Additional comments

My own text from the start of this review: “This work investigates if antipredator response
1) take place in woodlice (natural enemy vs 2 controls, which is nice set-up)
2) is different between sexes
3) vary according to the size of the woodlice
It discusses the differences in antipredator responses in evolutionary framework condisering costs and benefits of the behavior. " I reflect these points here:

The paper need to be a bit re-organised to make the set-up more understandable and some minor changes to reporting the results would be needed to better assess the results. I repeat what I said above:

- please mention breafly the 3 treatments in the Abstract and Aims
- please motivate the sex-difference approach in this species in the methods (are there results that made you believe that there could be sexual differences)
- please motivate the usage of size in the analyses in Aims or latest in the Methods

Further, please state more clearly if we know about the costs and about the benefits of the studied behavior in this animal. This could take place in Methods or earlier in Discussion. Now these information come a bit late in the discussion and the reader is getting doubts!

For the set-up, it is not clear from the data provided if all the woodlice got first the “rabbit” treatment. Lines 163-168.

If these questions can be answered (which they likely can be anwered), the ms makes a good addition to the study of sex-specific antipredator adaptations!

---

## Round 0.2 · Minor Revisions

Please re-revise to meet the concerns of Reviewer 2.

·

Basic reporting

I am satisfied with changes in manuscript - it is clear readible, fluent.

Experimental design

I am satisfied with changes in manuscript - experiment is described correctly and weaknesses are explained.

Validity of the findings

I am not sure about sex-based differences in results. Author tried to predict hypothesis based on weak findings from one paper (according to my opinion). In this corrected version he "softened" his results just by one sentence "this assumption can be considered plausible, but speculative".
On the other hand, I recognize his claim to be speculative - so I agree with such formulation.

Additional comments

on line 104 the is mistype - Horvátz beside Horváth

·

Basic reporting

I think that the story is very followable now.

Experimental design

All my questions are answered.

Validity of the findings

The findings are now better supported. I would still love to see pairwise comparison of the main factor "treatment". Supposingly this would indicate that the isopods respond in general to "toad" more than to "water". The pairwise comparisons between toad-rabbit-water would be important information for the first paragraph of discussion.

I asked for seeing the analyses of the first trial only: the first trial when the isopod experience any of the randomized treatments. I thought that the result would have been stronger for the first trial, but I failed with my assumption. This extra analyses is in appendices, but I think it is not needed.

In the same appendix, there is also an additional analyses: Full model without the factor "sex". I think that this is not needed for the "treatment" could be studied also from the full model. Further, there should be a good reason to leave out of the predictors - this could be done e.g. with AIC(c) comparisons between the models. BUT if "sex" is left out, then there is obviously no "sex x treatment" interaction, and that is an important parameter in this whole story.

I suggest a slightly different approach to the Conclusions to further deviate between what is actually found and what is speculation of the cost-benefit trade-off. The costs and benefits of rolling behaviour need still be studied, and thus the conclusions could concentrate on "predicting" how the cost and benefits should look like in males and females. In the annotated file there is a bad suggestion for such sentences (I commented to the file just as I would comment to my colleagues, hope this is ok).

Additional comments

We have a disagreement with the author about whether the significance treshold of 5 % should be corrected when conducting multiple pairwise comparisons. I leave it to the editor to decide if the multiple pairwise comparisons can be reported without Bonferroni correction or something alike. Nevertheless the main results would remain the same: the males with toad cue last longer time inactive than males with water treatment.

I think that the Fig. 1 should be done with the back-transformed values. Earlier there were the means and errors from the model, but in ln-scale. I hoped to see them back-transformed towards the raw data. Now the values are from the raw data, and theoretically this is not correct for we should see those values that the analyses actually use which are the ln-tranformed values - but... back-transformed. Now the Fig. 1 shows values that are not used in the analyses. My preference order is back-transformed values > ln-transformed values > raw data.

I tick the "minor revisions" for there are somethin reallyreally minor I suggested.

---

## Round 0.3 · accepted · Accept

Thank you for addressing all of the reviewers' concerns.